# Navigating Therapies, Challenges, and Recommendations for Treatment-Resistant Peripartum Depression: A Comprehensive Review

**DOI:** 10.3390/healthcare13192426

**Published:** 2025-09-25

**Authors:** Afshan Zeeshan Wasti, Sarah Almutairi, Mohsina Huq, Amal Hussain, Amal Mohamad Husein Mackawy, Farah Jabeen, Basmah Alharbi, Anjuman Gul Memon, Mawahib Ahmed

**Affiliations:** 1Department of Medical Laboratories, College of Applied Medical Sciences, Qassim University, Buraydah 51452, Saudi Arabia; sarahsaad@gmail.com (S.A.); m.huq@qu.edu.sa (M.H.); ama.ali@qu.edu.sa (A.H.); mkaoy@qu.edu.sa (A.M.H.M.); 2Department of Biochemistry, Jinnah University for Women, Karachi 74600, Pakistan; dr.farahjabeen@juw.edu.pk; 3Department of Basic Health Sciences, College of Applied Medical Sciences, Qassim University, Buraydah 51452, Saudi Arabia; b.alwahbi@qu.edu.sa; 4Department of Biochemistry, College of Medicine, Qassim University, Buraydah 51452, Saudi Arabia; a.memon@qu.edu.sa

**Keywords:** treatment-resistant depression, peripartum depression, pharmacological and non-pharmacological interventions, pharmacotherapy, psychology, neuromodulation, review

## Abstract

Treatment-resistant peripartum depression (TRPD) is a significant public health concern due to the dual imperative of maternal symptom relief and fetal/neonatal safety with complex therapeutic challenges, particularly among expecting mothers worldwide. This comprehensive review focused on current pharmacological and non-pharmacological interventions for treatment-resistant depression (TRD)/peripartum depression (PPD), highlighting their mechanisms, efficacy, safety profiles, and practical considerations. The search strategy is based on PRISMA (Preferred Reporting Items for Systematic Reviews and Meta-Analysis), using a systematic search of PubMed, Cochrane Library, and EMBASE for English-language articles published between 2000 and 2024, with a combination of Medical Subject Headings (MeSH) and free-text terms for TRD/TRPD. After screening, the initial search yielded 142 articles; only 67 articles were qualified for eligibility and quality assessment. According to related research, pharmacological treatments such as SSRIs or brexanolone and zuranolone can be effective in addressing TRPD challenges, but they carry concerns regarding fetal and neonatal risk. In contrast, non-pharmacological interventions—such as cognitive behavioral therapy (CBT), repetitive transcranial magnetic stimulation (rTMS), and exercise—offer safe, evidence-based alternatives that are becoming increasingly accessible. Our findings imply that innovative therapeutics and integration of these interventions personalized to individual needs are the optimal clinical approach that may help in balancing maternal symptom control and perinatal safety. Also, expanded mental health infrastructure with enhanced research is essential for advancing TRPD care.

## 1. Introduction

Peripartum depression (PPD) affects 15–20% of expecting mothers worldwide in the months following childbirth, also called postpartum depression. Effective management of PPD is not only a clinical priority due to profound implications for maternal, fetal, and infant health but also a public health imperative [1,2]. To discriminate between different interlinked terminologies clearly, it is evident that peripartum depression (PPD) is a subtype of major depressive disorder (MDD) occurring during pregnancy or within 12 months postpartum. MDD is characterized by persistent low moods, anhedonia, exhaustion, and cognitive deficits lasting at least two weeks. Treatment-resistant depression (TRD) is defined as a major depressive episode that fails to respond to at least two adequate antidepressant trials at an adequate dose and duration, consistent with established TRD criteria. TRD significantly impairs maternal health, child development, and family functioning. Treatment-resistant peripartum depression (TRPD) is diagnosed when the challenges of both PPD and TRD occur during the peripartum period [3,4].

Despite multiple antidepressant trials, reported studies observed that up to 40% of peripartum depressed women may not respond to the initial therapy, while up to 60% do not achieve remission [4,5], and these figures may indicate a pressing need to better understand and manage TRD. Treatment-resistant peripartum depression (TRPD) is defined as peripartum depression that fails to remit after at least two adequate trials of antidepressant therapy, consistent with established TRD criteria [6]. TRPD can severely impact a mother’s physical and mental health and ability to form strong attachments to her child, leading to a negative impact on a child’s health, increasing the risk of behavioral problems, cognitive challenges, and developmental delays [7,8,9]. The condition may exhibit a variety of typical symptoms, including consistent sorrow, loss of interest in activities, exhaustion, impatience, changes in appetite, sleep troubles, and concentration difficulties and are linked to certain conditions, according to the DSM-V criteria [10]. In addition, anxiety, mood swings, and suicidal thoughts may occur frequently in individuals with TRPD [11,12].

This review focuses exclusively on maternal treatment-resistant peripartum depression (TRPD), as most available epidemiological data and clinical trials address women, reflecting both the higher prevalence of peripartum depression in mothers and substantial gaps in paternal TRPD research. While paternal peripartum depression affects approximately 10% of fathers, the literature on treatment resistance, unique risk factors, and specific interventions in fathers remains limited due to pervasive stigma and under-recognition in clinical settings. Future reviews should consider this unmet need as this evidence base grows.

### 1.1. Associated Factors/Comorbidities

A complex interaction of genetic, neurobiological, and psychosocial factors is linked with TRD. Genetic factors have been found to influence both a person’s chance of getting depression and how likely they are to respond to treatment. During the peripartum period, hormonal fluctuations, particularly in estrogen and progesterone, have been associated with increased risk for depression and may contribute to treatment resistance [13].

While psychosocial factors contribute to peripartum depression and may increase TRPD risk, biological mechanisms—such as altered serotonin, dopamine, and norepinephrine neurotransmitter levels, and HPA axis dysregulation—are also implicated in its pathophysiology [4,14].

TRD is frequently combined with other mental health conditions, such as anxiety, post-traumatic stress disorder, and substance use disorders. These comorbidities complicate diagnosis and treatment, and are associated with increased risks of cardiovascular disease, reduced quality of life, and functional impairment. Additionally, medical conditions like thyroid dysfunction and chronic pain contribute to reduced productivity and higher healthcare costs, which may further influence the incidence and severity of TRPD [4,14].

### 1.2. Diagnostics Obstacles

Accurate diagnosis remains challenging due to the complex and varied nature of TRD in individuals with peripartum depression, and its overlap with other mental health problems. Persistent depressive symptoms despite at least two adequate antidepressant trials, in the absence of reliable biomarkers, complicate clinical decision-making [4].

It is important to note that a considerable proportion of evidence for TRPD is extrapolated from general TRD studies. Due to physiological, hormonal, and psychosocial differences in the peripartum period, the generalizability of TRD findings to TRPD may be limited. Dedicated perinatal research is urgently needed.

The present review highlights the current challenges and emerging evidence, with critical assessment of the efficacy and limitations of both pharmacological and non-pharmacological interventions in treatment-resistant depression (TRD) and peripartum depression (PPD), to inform clinical practice and guide future research to better support women experiencing treatment-resistant peripartum depression (TRPD).

## 2. Methods

In the present review, the search strategy is based on PRISMA-Preferred Reporting Items for Systematic Reviews and Meta-Analysis criteria [15] to evaluate the current and emerging therapies for individuals with treatment-resistant peripartum depression.

The inclusion criteria were based on a systematic search of PubMed, Cochrane Library, and EMBASE for English-language articles published between 2000 and 2024. The review focused on (*applied filters*: abstract, full text, meta-analysis, systematic review) with a combination of Medical Subject Headings (MeSH) and free-text terms were used, including (“treatment resistant depression” OR “TRD”) AND (“peripartum depression” OR “postpartum depression” OR “perinatal depression” OR “PPD” OR “treatment resistant peripartum depression” OR “TRPD”) AND (“therapy” OR “treatment” OR “intervention”). Search terms included “peripartum depression”, “postpartum depression”, and “perinatal depression” were not entered as exact phrases to maximize inclusion of international studies and nomenclature, but as Boolean combinations to maximize sensitivity and capture relevant variations in terminology.

Studies were limited by exclusion criteria, including studies not related to peripartum or postpartum depression, outcomes not attributable to women, not a quantitative study, not meeting criteria for treatment resistance (TRD/PPD/TRPD), not published in a peer-reviewed journal, published in another language, and having insufficient information. However, we restricted our inclusion to studies on maternal TRPD to ensure consistency in the population studied and due to the scarcity of high-quality research on paternal TRPD. Likewise, while child developmental outcomes are mentioned in context, our primary focus remains on maternal outcomes and interventions, as dictated by the scope and prevalence of current evidence.

After screening, the initial search yielded 142 articles; only 67 articles were qualified for eligibility. Although this number may limit the generalizability of some conclusions, it provides an analytically manageable and clinically meaningful dataset for review. A larger set may have introduced redundancies or diluted specificity. Most current PPD/TRPD studies are limited by small sample sizes, heterogeneity in patient populations, and short-term follow-up; this constrains the generalizability of findings and warrants further large-scale, long-term trials. The quality assessment was performed using the Newcastle–Ottawa Scale for observational studies and the Cochrane Risk of Bias tool for randomized trials. Of the 67 included studies, quality appraisal analyses found that 38 were at low risk of bias, 20 at moderate risk, and 9 at high risk. High-quality studies were prioritized in the synthesis. With two independent reviewers’ assessments, discrepancies were resolved by consensus (Figure 1).

### 2.1. TRPD Current Therapeutics

Numerous interventions were evaluated in the previous studies, including pharmacotherapy, psychotherapy, and neuro-modulation techniques. Below, we provide a more detailed synthesis of the evidence for each intervention, with emphasis on effectiveness and limitations in the peripartum context. Table 1 and Table 2 summarize the targeted pharmacological and non-pharmacological interventions for TRD and PPD, with details of available drugs, mode of action, and outcomes where available. Table 3 depicts the integrated comparison with various fundamental aspects of these interventions.

### 2.2. Pharmacological Interventions

Pharmacological interventions for TRD/PPD include switching to a different antidepressant, augmenting current medication, or combining multiple antidepressants. SSRIs are commonly used as first-line pharmacological therapy for peripartum depression, but evidence for their effectiveness in TRPD is mixed. Some studies report modest improvements, but remission rates remain low in this population [6]. Intranasal esketamine [16], immunomodulators [17], and glutamate modulators [18] have been explored in small trials, but data specific to peripartum populations are limited.

The main pharmacological bridge between TRD and PPD is SSRIs and SNRIs, as demonstrated in Table 1, suggesting that, with sertraline considered safest in lactation, brexanolone and zuranolone are unique to PPD [19,20]. The other drugs, such as ketamine/esketamine, show rapid efficacy, while omega-3s are low-risk, evidence-supported adjuncts in both TRPD, especially for severe cases having inflammation-linked symptoms. Emerging evidence indicates that a subset of severe TRPD cases exhibits elevated inflammatory markers (e.g., C-reactive protein, interleukin-6), suggesting an inflammation-linked endophenotype. This raises the prospect of targeted immunomodulatory or anti-inflammatory therapies, though further research is needed to clarify treatment response and the impact of perinatal hormonal-immune changes on inflammation-driven depression [20]. Moreover, several agents like MAOIs, TCAs, and atypical antipsychotics are effective in TRD but are not suitable during the peripartum period due to safety concerns.

### 2.3. Augmentation Strategies

Augmentation Strategies involve adding a second medication to an existing antidepressant regimen. Several medications, such as lithium and atypical antipsychotics, have been investigated as potential augmenting agents in TRD patients [21]. Switching to a different medication and adding adjunctive therapies were associated with a higher likelihood of remission in TRD than augmenting or combining medications [22]. However, there is little data to back up medication used for TRD [6], but most studies excluded peripartum populations, limiting generalizability [23]. Overall, there is a paucity of high-quality evidence specific to TRPD, and most recommendations are extrapolated from general TRD populations.

The other treatment option is the use of adjunctive drugs for non-response or partial response to an antidepressant, including Aripiprazole, Quetiapine, and Risperidone as a first line, and Brexpiprazole, Bupropion, Lithium, Mirtazapine/Mianserin, Modafinil, Olanzapine, Triiodothyronine as a second line, and Esketamine for TRD [24]. A rational pharmacologic approach for TRPD includes reevaluating diagnosis, assessing prior treatment response and tolerability, using adjunctive medications judiciously, discontinuing ineffective medications, and closely monitoring symptoms and adverse effects [25].

**Table 1 healthcare-13-02426-t001:** Targeted pharmacological interventions for TRD and peripartum depression (PPD).

Category	Agent/Class	Mechanism of Action/Modality	Level of Evidence (TRD)	Level of Evidence (PPD)	Main Safety Warnings (Perinatal)	References
First-line Antidepressants	SSRIs (e.g., fluoxetine, sertraline)	Selective serotonin reuptake inhibition	Common first-line; often insufficient alone in TRD	Widely used; sertraline preferred in breastfeeding	Good safety profile; low infant exposure via breast milk	[26,27]
	SNRIs (e.g., venlafaxine, duloxetine)	Serotonin and norepinephrine reuptake inhibition	Used when SSRIs fail; moderate efficacy	Used cautiously postpartum	Venlafaxine frequently used postpartum; monitor for side effects	[26,27]
	Bupropion	Dopamine–norepinephrine reuptake inhibition	Alternative option; especially for fatigue, anhedonia	Limited use in breastfeeding	Limited data in lactation; caution advised	[26]
	Mirtazapine	Noradrenergic and specific serotonergic action	Often used as augmentation; sedation useful in insomnia	Limited safety data in lactation	Consider if insomnia or appetite loss dominate	[26]
	Vortioxetine	Multimodal serotonergic activity	Modest efficacy in TRD; well tolerated	Not commonly used in PPD	Cognitive benefit in TRD	[26,27]
	Agomelatine	Melatonergic agonist and 5-HT2C antagonist	Mild-moderate benefit in TRD	Limited data in PPD	Non-sedative; supports circadian rhythm	[26]
Second-line/Other Antidepressants	TCAs (e.g., amitriptyline)	Serotonin and norepinephrine reuptake inhibition	Used after SSRI/SNRI failure	Not typically used in PPD	Side effect burden, especially anticholinergic	[28]
	Quetiapine, trazodone	Multi-receptor activity	Used for augmentation; quetiapine has evidence in TRD	Not recommended in PPD due to sedation	Monitor metabolic and sedative side effects	[28]
	Vilazodone, levomilnacipran	Serotonergic and norepinephrine/dopamine action	Considered after SSRI failure	Limited to no use in PPD	Novel agents, limited perinatal data	[28]
MAO Inhibitors	Phenelzine, tranylcypromine, etc.	Inhibit MAO-A/B, prevent monoamine breakdown	High efficacy in atypical/TRD	Contraindicated in pregnancy/lactation	Dietary restrictions and hypertensive crisis risk	[29]
Neurosteroid	Brexanolone (IV allopregnanolone)	Positive allosteric modulator of GABA-A receptors	Not indicated for TRD	FDA-approved for PPD	Rapid action; 60-hour IV infusion required	[19]
	Zuranolone (zurzuvae) oral neurosteroid	positive allosteric modulator of GABA-A receptor)	Not indicated for TRD	FDA (2023) approved for PPD	Oral alternative to brexanolone’s IV infusion, with evidence for rapid, sustained symptom control from multiple RCTs.	[20]
NMDA Receptor Modulators	Esketamine (intranasal)	NMDA receptor antagonist; enhances glutamate signaling	FDA-approved for TRD; rapid effect	Not approved in pregnancy; possible postpartum use	In-clinic monitoring due to dissociation	[16]
	Ketamine (IV)	Non-selective NMDA receptor antagonist	Effective in 60–70% of TRD cases	Experimental use in severe PPD	Short-acting; some pilot postpartum trials	[18]
	(2R,6R)-HNK	Ketamine metabolite; NMDA-independent	Promising early trials; non-dissociative	Not yet studied in PPD	Still investigational	[30]
Augmentation Strategies	Lithium, atypical antipsychotics	Enhance monoamine function; antipsychotic modulation	Effective as augmentation in TRD	Lithium not preferred during breastfeeding	Requires serum monitoring; side effect risk	[21]
Immunomodulators	Infliximab, tocilizumab	Anti-cytokine (TNF-α, IL-6) inhibition	Promising in inflammation-related TRD	Not approved or studied in PPD	Off-label; biomarker-guided therapy	[17]
Anti-inflammatory Agents	NSAIDs (e.g., celecoxib), cytokine blockers	Inhibit peripheral and CNS inflammation	Moderate benefit in inflamed TRD cases	Limited peripartum safety data	GI and cardiovascular risks; adjunctive only	[31]
Psychedelic-Assisted Therapy	Psilocybin, MDMA	5-HT2A receptor agonism, neuroplasticity facilitation	Rapid effects in trials for TRD	Not studied or approved for PPD	Requires psychotherapeutic setting	[32]
Cannabinoids	CBD (cannabidiol)	Endocannabinoid modulation, serotonin activity	Preliminary TRD evidence	Insufficient safety data for PPD	Not FDA-approved; formulation challenges	[33]
Novel Agents	Kappa opioid antagonists, sigma-1 modulators, etc.	Non-monoaminergic targets	Early-phase trials in TRD	No data in PPD	Potential future treatments	[34]
Nutritional Adjuncts	Omega-3 fatty acids (EPA/DHA)	Anti-inflammatory; membrane fluidity; serotonin modulation	Safe adjunct in TRD	Safe during pregnancy and lactation	Modest efficacy; neurodevelopmental benefit in PPD	[35]

Legend: TRD: treatment-resistant depression; PPD: peripartum depression; SSRIs/SNRIs: selective/serotonin–norepinephrine reuptake inhibitors; NMDA: N-methyl-D-aspartate; MAO: monoamine Oxidase; EPA/DHA: eicosapentaenoic acid/docosahexaenoic acid. Level of Evidence (TRD/PPD) indicates, e.g., “High (multiple RCTs, meta-analyses)/Low (case reports)” or “Low—Not recommended for peripartum use”. Main Safety Warnings (perinatal) (e.g., contraindicated in pregnancy; risk of neonatal withdrawal; limited lactation safety). Note: For each drug/class, despite strong evidence, requires close monitoring in peripartum.

### 2.4. Non-Pharmacological Interventions

Non-pharmacological interventions for TRD include psychotherapy, electroconvulsive therapy (ECT), and transcranial magnetic stimulation (TMS). ECT has demonstrated response rates of 60–70% in severe TRD, including peripartum cases, but stigma and side effects limit its use [36]. Digital therapeutics [37] and virtual reality therapy [38] are evidence-based therapies potentially effective in treating depression, including TRD. Smartphones and other electronic devices can access software-based interventions, or “digital therapeutics,” explicitly designed to treat depression and other mental health conditions. TMS and digital therapeutics have shown promise in non-peripartum TRD, but data in peripartum populations are sparse and further research is needed [39]. Virtual reality therapy and neurofeedback are emerging modalities with limited but growing evidence.

Psychotherapy, particularly cognitive-behavioral therapy (CBT) and interpersonal therapy (IPT), is recommended for TRPD, especially in individuals with a history of trauma or comorbid anxiety disorders. However, there is insufficient evidence to support psychotherapy as a stand-alone treatment for TRPD; most studies suggest benefit as part of a multimodal approach. Lifestyle modifications such as regular exercise, a healthy diet, and stress management may provide additional benefit [40,41].

**Table 2 healthcare-13-02426-t002:** Integrated non-pharmacological interventions for TRD and peripartum depression (PPD).

Intervention	Modality/Mechanism	Evidence and Efficacy	Level of Evidence (TRD)	Level of Evidence (PPD)	Main Safety Warnings (Perinatal)	References
Cognitive Behavioral Therapy (CBT) (including CBTi)	Psychotherapy to restructure thoughts and behaviors; CBTi targets insomnia	Strong RCT/meta-analysis support; effective monotherapy or adjunct in TRD and PPD	Widely used and effective	Strong efficacy in pregnancy and postpartum	Safe during pregnancy/lactation; flexible delivery (teletherapy, group, individual)	[42,43]
Mindfulness-Based Cognitive Therapy (MBCT)	Combines CBT with mindfulness to reduce rumination	Evidence supports relapse prevention in TRD	Effective in residual symptoms, relapse prevention	Limited data, but likely safe and beneficial	Gentle, non-invasive, supports emotional regulation	[44]
Digital Therapeutics	App-based CBT and mood regulation tools	Growing evidence for mild/moderate TRD	Beneficial as adjunct or for access-limited patients	Under investigation for PPD	High accessibility; digital literacy dependent	[37]
Virtual Reality Therapy	Immersive environments for therapeutic exposure	Emerging data in TRD, especially PTSD comorbidity	Promising in emotional regulation	Not yet studied in PPD	Costly and niche; not first-line	[38]
Electroconvulsive Therapy (ECT)	Induces seizure via electrical stimulation	Gold standard in severe TRD (50–70% response)	Widely used in refractory TRD	Rarely used in PPD (case-based)	Not first-line in peripartum due to anesthesia/memory concerns	[36]
Bifrontal ECT	Modified ECT with fewer cognitive side effects	Comparable efficacy to bilateral ECT	Considered safer cognitively in TRD	No data in PPD	Still requires anesthesia	[42]
Repetitive Transcranial Magnetic Stimulation (rTMS)	Magnetic pulses stimulate dorsolateral prefrontal cortex	~50% response in TRD; FDA-approved	Widely used in TRD	Demonstrated safety and efficacy in postpartum depression	Medication-free; non-invasive; safe during lactation	[45,46,47]
Deep TMS (rdTMS)	Reaches deeper cortical and subcortical areas	Some evidence of superior efficacy in TRD	Effective in TRD	Not yet validated for PPD	High cost, limited availability	[48,49]
tDCS (Transcranial Direct Current Stimulation)	Low current alters cortical excitability	30–40% response in TRD	Used as low-cost, portable option	Limited data in PPD	Theoretical safety; not mainstream in perinatal settings	[50]
Deep Brain Stimulation (DBS)	Implanted electrodes target deep brain regions	Up to 60% response in severe, refractory TRD	Experimental in advanced TRD	Not applicable in PPD	Invasive; contraindicated in pregnancy	[51]
Neurofeedback	Brainwave training via real-time feedback	Promising early TRD studies	Potential for self-regulation support	No current data for PPD	Training-intensive; not routine	[52]
Exercise/Physical Activity	Enhances BDNF, endorphins, circadian rhythm	Strong efficacy in mild-moderate depression; adjunct in TRD	Effective adjunct in TRD	Reduces PPD symptoms (SMD −0.41 to −0.53)	Safe, low-cost, improves physical/mental health	[53,54,55]
Mind-Body Practices (e.g., Yoga)	Integrates movement, breathing, and mindfulness	Beneficial in both TRD and PPD	Used as adjunctive support	Particularly effective postpartum	Improves mood, sleep, and bonding	[56]
Peer/Social Support and Psychoeducation	Group sessions, home visits, non-specialist support	Strong community-based evidence in PPD	Limited evidence in TRD; emerging in group therapy	Effective in reducing PPD, especially in low-resource settings	Safe, culturally adaptable, scalable	[57]
Gene Therapy	Modifies gene expression via viral vectors	Preclinical success in TRD models	Experimental stage in TRD	No use in PPD	Ethical issues; not ready for clinical use	[58]

Table 2 explains the double comparison between non-pharmacological interventions effective in both TRD and PPD, the CBT and exercise. Efficacious with high safety in perinatal contexts, while rTMS shows strong dual-utility, and is FDA-approved for TRD and supported by meta-analyses in PPD. Peer support is uniquely potent in PPD, especially in low-resource or culturally sensitive contexts; however, several invasive or high-tech interventions (DBS, ECT, VR) are TRD-specific and not applicable to peripartum populations due to safety or practicality. Additionally, Lifestyle interventions such as exercise and yoga offer low-cost, high-accessibility options particularly suited for postpartum recovery and mood support in TRPD.

Pharmacological and non-pharmacological interventions with TRD and PPD in comparison with different aspects like treatment modalities and mechanisms, efficacy and evidence base, safety and side effects, and practical considerations were illustrated in Table 3. It suggests that the pharmacological options remain foundational, especially in moderate to severe cases of TRD and PPD but are often limited due to safety concerns during pregnancy and lactation in PPD. Multimodal strategies and frequent augmentation are the basic requirements for the TRD treatment, while early detection and individualization with low-risk support systems helped PPD interventions. However, non-pharmacological approaches offer safe, effective, and scalable alternatives, especially important in TRPD settings where drug-free treatments are preferred.

**Table 3 healthcare-13-02426-t003:** Integrated comparison of pharmacological vs. non-pharmacological interventions in TRD and PPD.

	Aspect	Pharmacological Interventions	Non-Pharmacological Interventions
1.	Treatment Modalities and Mechanisms	TRD: Targets monoamine systems (serotonin, norepinephrine, dopamine), glutamate (e.g., ketamine), neurosteroids, inflammation. Often biochemical. PPD: Includes SSRIs, SNRIs, brexanolone, zuranolone (GABA-A modulator), with focus on safety in pregnancy/lactation.	TRD: Includes neuromodulation (ECT, TMS, DBS), psychotherapy (CBT, MBCT), and lifestyle/digital tools. Aims to restore neuroplasticity, cognition, emotional processing. PPD: Includes CBT/MBCT, rTMS, exercise, peer support, and digital therapies—non-invasive and safe for perinatal populations.
2.	Efficacy and Evidence Base	TRD: Esketamine and ketamine offer rapid symptom relief. Antidepressants often need augmentation. MAOIs and glutamatergic agents show promise. Broad evidence base. PPD: SSRIs, brexanolone, and zuranolone supported by RCTs. Offers rapid onset but is complex to administer. Concerns about breastfeeding and fetal safety persist.	TRD: ECT and rTMS show robust short-term results; CBT/MBCT support long-term remission; digital tools are emerging. PPD: CBT and MBCT are highly effective; rTMS has growing support; exercise and peer-based support reduce symptoms and promote maternal-infant bonding.
3.	Safety and Side Effects	TRD: Risk of sedation, metabolic and cardiovascular side effects; ketamine has misuse potential. Requires monitoring (e.g., lithium levels, MAOI diets). PPD: Risk of teratogenicity and neonatal effects; brexanolone requires 60-hour infusion under supervision. Breastfeeding exposure is a concern.	TRD: Side effects include cognitive effects (ECT), discomfort (TMS), but minimal systemic toxicity. PPD: Very safe—no systemic drug exposure. rTMS and CBT are well tolerated. Exercise low-risk when guided. No known adverse effects on fetus or infant.
4.	Practical Considerations	TRD: Widely available agents; some novel treatments are expensive (e.g., esketamine). Pharmacogenomics in early use. PPD: SSRIs are first-line but patient hesitancy is common. Brexanolone is costly, IV-only, and underutilized.	TRD: Access limited by cost/availability (TMS, DBS); therapy requires time and engagement. Long-term cost-effective. PPD: Access challenges for TMS and therapy in some areas; digital and behavioral options increase scalability. High patient acceptance due to safety.

### 2.5. Critical Appraisal

This review offers valuable pathways toward personalized and safe care for peripartum depression, highlighting that both pharmacological and non-pharmacological interventions each with unique strengths and limitations.

While pharmacological therapies remain foundational, especially for moderate to severe TRPD, they are often constrained due to fetal or neonatal safety concerns [12]. Despite SSRIs like sertraline having a well-established safety profile during lactation and significant efficacy in moderate to severe cases [59]. Being widely used off-label for PPD with favorable safety data, the absence of FDA approval can lead to insurance restrictions and varying levels of acceptance among clinicians and patients. This drug has strong evidence for efficacy in PPD but is not specifically FDA-approved for PPD; caution is warranted and individual risk–benefit must be considered. Conversely, FDA-approved agents like brexanolone and zuranolone increase access and clinician confidence but face challenges related to cost and delivery (e.g., brexanolone’s 60-hour infusion requirement).

MAOIs have high efficacy in TRD but are not recommended in perinatal populations due to significant safety concerns. Caution is advised with other antidepressants, and shared decision-making is critical, especially during pregnancy, where the balance between maternal benefits and fetal risks must be carefully navigated [60].

Recently, the role of GABA–glutamate dysregulation has emerged as a key mechanism in TRD and TRPD, particularly through modulation of GABA-A receptors. Novel agents like brexanolone and zuranolone, a GABA-A receptor modulator and ketamine analogs specifically approved for PPD, have demonstrated rapid, sustained symptom relief, though access and cost remain barriers [19]. It provides a more practical option for patients while maintaining robust efficacy in improving depressive symptoms within days, and may represent a breakthrough for women who do not respond to traditional antidepressants [19,20].

In contrast, non-pharmacological interventions—notably psychotherapy modalities such as cognitive behavioural therapy (CBT) and mindfulness-based cognitive therapy (MBCT), neurostimulation techniques including repetitive transcranial magnetic stimulation (rTMS), and adjunctive lifestyle interventions such as structured exercise—offer safe, effective, and increasingly accessible alternatives or complements to pharmacotherapy.

These approaches avoid systemic drug exposure and align well with patient preferences for non-medication options, especially during pregnancy and breastfeeding. However, barriers including limited availability, treatment adherence challenges, and costs persist, underscoring the need for expanded infrastructure and integration of digital therapeutics to improve reach. Meta-analyses support CBT and MBCT as highly efficacious for reducing depressive symptoms, while rTMS and physical activity show moderate to strong benefits with minimal risk [57,61,62,63,64,65].

The limited inclusion of peripartum women in clinical trials has left a significant gap in safety data. International bodies are now calling for better-designed studies and registries to ensure evidence-based pharmacological care in this vulnerable population. Personalizing pharmacological strategies in TRPD requires careful assessment of prior treatment responses, safety profiles during pregnancy and lactation, and close symptom monitoring. More studies are needed to evaluate the safety and effectiveness of these treatments; for those who have not reacted to conventional antidepressant therapy, the fact that so many alternative therapies exist is reassuring that there may still be hope.

### 2.6. Challenges and Prospects

The preceding sections provided an integrated, section-by-section critique of pharmacological, augmentation, and non-pharmacological strategies, while this section highlights overarching challenges across all approaches.

Treatment-resistant peripartum depression (TRPD) is the worldwide health issue impact people from various cultures and geographical areas, particularly complex for expecting mothers and has significant challenges besides other genders, like common fluctuations in hormonal levels and/or comorbid conditions and family and social pressures, inhibition of willingness to seek treatment, and further exploitation to worsen the depressive symptoms and complicate the treatment of TRPD [66,67].

It is assumed that in clinical practice, the most realistic and effective approach to TRPD is a personalized, multidisciplinary treatment model that judiciously integrates pharmacological and non-pharmacological strategies. This model prioritizes symptom severity, patient safety, and preference while leveraging the complementary strengths of rapid symptom control via medication and sustainable remission supported by psychotherapy and neurostimulation.

Long-standing depression, variability in treatment response, financial stress, personal and social issues, and women’s reproductive health are other contributing factors that could further complicate the issue, and it may require more intensive and multifaceted treatment approaches to address these challenges. Looking forward, ongoing research into novel agents with improved safety profiles, refinement of neurostimulation protocols, and scalable digital therapeutics hold promises for advancing TRPD care. Equally important is the enhancement of the perinatal mental health services infrastructure to ensure equitable access to comprehensive treatment options.

### 2.7. Recommendations for TRPD

Early identification of TRPD, close monitoring, and collaborative decision-making with patients are paramount to optimizing outcomes. Strengths of the current literature include growing recognition of the need for tailored approaches in peripartum populations and emerging evidence for novel interventions (e.g., digital therapeutics, neuromodulation). However, weaknesses include small sample sizes, lack of standardized definitions for TRPD, and limited long-term outcome data.

Individual treatment decisions should consider the patient’s preferences, risks, and potential advantages of using pharmacological or non-pharmacological interventions for TRD/PPD in women. Future studies should work to deepen our comprehension of the root causes and risk factors for TRD/PPD and create secure and efficient treatment alternatives for the women at risk for TRPD and offer prompt and effective alternative therapies; raising awareness and enhancing screening and diagnostic methods is vital. To improve outcomes for this vulnerable population, researchers and clinicians are urged by this critical review to give alternate treatments for TRPD in women with peripartum depression top priority.

An interdisciplinary approach is added in the recommendations involving the team of experts, which may include psychiatrists, psychologists, social workers, nutritionists, and other specialists to offer the most appropriate treatment for TRPD patients according to the individual’s needs, with an evidence-based treatment plan.

Furthermore, adopting and adapting effective framework models (e.g., the “Perinatal Mental Health Community Services” model-UK; *MumSpace* model perinatal depression and anxiety—Australia), that demonstrate investment in multidisciplinary perinatal mental health teams, universal screening, integrated care pathways, and digital mental health solutions can dramatically improve maternal outcomes and service access. Using such proven models in local health systems should be prioritized in future policy and practice initiatives [68,69].

In summary, the future of PPD/TRPD care lies not in choosing between pharmacological and non-pharmacological interventions, but in strategically combining them to support each woman’s unique needs, values, and life circumstances. Continued research, education, and policy-level commitment are essential to closing the treatment gap and enhancing the mental health of mothers and their children.

## 3. Conclusions

Treatment-resistant peripartum depression (TRPD) poses a significant clinical challenge due to the complex interplay of maternal physiological changes, fetal safety considerations, and the profound psychosocial impact of untreated maternal mental illness. This review examines various contributing factors, challenges, and recommendations that highlight the potential interventions for the TRD/PPD, which offers a substantial therapeutic issue in women with limited evidence-based treatment options and considerable negative impacts on mother and child health outcomes.

Comparison of the relative effectiveness of pharmacological interventions like SSRIs, brexanolone, and zuranolone demonstrates moderate-to-strong evidence for symptom reduction in TRPD (particularly moderate-to-severe cases), while non-pharmacological methods such as CBT and rTMS also show robust efficacy and often superior safety/tolerability profiles in perinatal women. Meta-analyses suggest that the magnitude of benefit for CBT in mild-to-moderate PPD may equal or exceed that of some pharmacological agents, and combined therapy is recommended for the most resistant cases.

Our findings imply that no one treatment works for all cases of TRPD and that a combination of therapy and an interdisciplinary approach with a team of experts may be the most successful strategy.

Future research should prioritize randomized controlled trials specifically enrolling peripartum individuals with TRD; standardized definitions of treatment resistance and head-to-head comparisons of interventions, with a critical lens on their feasibility, efficacy, and safety within the peripartum context; and also address safety data gaps by including pregnant and lactating participants in clinical trials.

## Figures and Tables

**Figure 1 healthcare-13-02426-f001:**
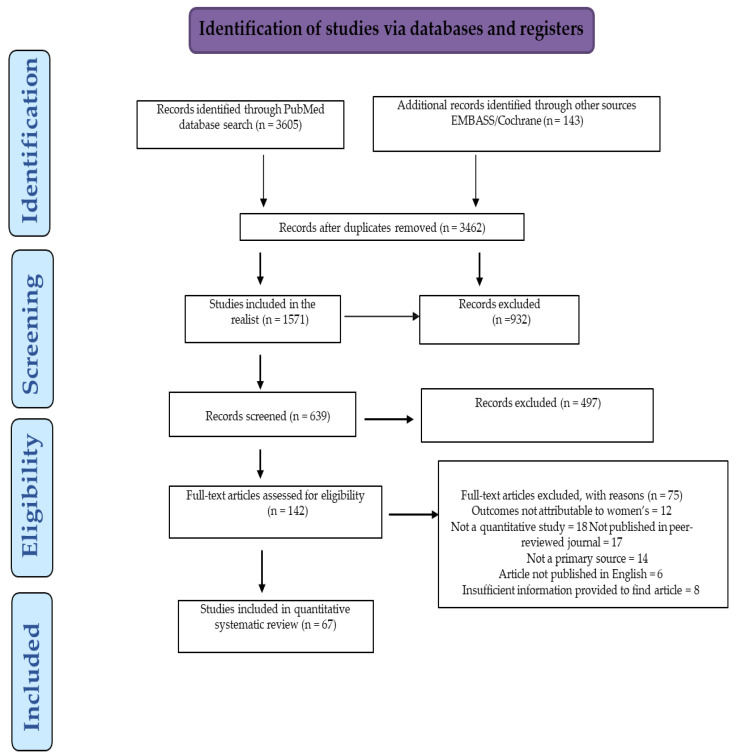
The mechanism for selecting records related to the study according to the PRISMA guidelines.

## Data Availability

The data used to support the findings of this study are included within the article. Should any raw data files be needed in another format, they are available from the corresponding authors upon reasonable request.

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
