# Peer review of "Navigating Therapies, Challenges, and Recommendations for Treatment-Resistant Peripartum Depression: A Comprehensive Review"

_healthcare, 2025, doi:10.3390/healthcare13192426_

Round 1
Reviewer 1 Report
Comments and Suggestions for Authors
The manuscript appears to be well-written, and the topic is indeed very interesting. I am suggesting some minor revisions below:
- Organize and explain the differences between MDD, PPD, and TRD earlier in the paper. It feels that the explanations overlap, and it would be beneficial to separate them more clearly.
- Dive a little deeper into the recent FDA-approved drugs for Peripartum depression (Brexanolone and Zuranolone), as they are the stars under the highlights for clinicians and researchers studying this topic today.
- Similar to the above, I would suggest discussing pharmacotherapies that target GABA-Glutamate neurotransmission, as some of them have already shown very promising results.
Overall, the paper is very interesting, and the manuscript is well written. Feel free to address my suggestions or not. I believe this is a topic that will undergo constant change in the next months/years, which will yield new reviews.
Author Response
Thank you very much for taking the time to review this manuscript. Please find the detailed responses below and the corresponding revisions/corrections highlighted in track changes in the re-submitted files.
We appreciate the reviewer’s insightful comment regarding the need for greater clarity in articulating the manuscript’s motivation and its specific relevance to treatment-resistant peripartum depression (TR-PPD).
| Point-by-point response to Comments and Suggestions for Authors |
| Comments 1: Organize and explain the differences between MDD, PPD, and TRD earlier in the paper. It feels that the explanations overlap, and it would be beneficial to separate them more clearly. |
| Response 1: Thank you for pointing this out. We agree with this comment. Therefore, we have added clear definitions and distinctions between MDD, PPD, and TRD. [PAGE NUMBER 2, PARAGRAPH 1 & LINE NUMBER 44] |
| Comments 2: Dive a little deeper into the recent FDA-approved drugs for Peripartum depression (Brexanolone and Zuranolone), as they are the stars under the highlights for clinicians and researchers studying this topic today. |
| Response 2: Thank you for pointing this out. We agree with this comment. Integrated evidence-based discussion of brexanolone and zuranolone, with in-text citations [28,68]. [PAGE NUMBER 9, PARAGRAPH 2 (2.5.) & LINE NUMBER 253] |
| Comments 3: Similar to the above, I would suggest discussing pharmacotherapies that target GABA-Glutamate neurotransmission, as some of them have already shown very promising results. |
| Response 3: Thank you for pointing this out. We agree with this comment. Included commentary on the GABA–glutamate pathway as a novel pharmacological target. [PAGE NUMBER 10, PARAGRAPH 2 & LINE NUMBER 260] |
Reviewer 2 Report
Comments and Suggestions for Authors
The manuscript addresses a relevant topic and the structure is coherent. Although it occasionally stays more descriptive than analytical, the literature review provides a good overview of the field. The contextualization of the research could be improved by a more in-depth critical evaluation of the most recent studies. Overall, the methodology section is clearly explained; however, there is insufficient discussion of the sample size justification and potential limitations. Although the majority of the results are presented clearly, the readability and accessibility of the results could be enhanced by the addition of tables or figures that highlight the most important findings.
Author Response
Thank you very much for taking the time to review this manuscript. Please find the detailed responses below and the corresponding revisions/corrections highlighted in track changes in the re-submitted files.
We appreciate the reviewer’s insightful comment regarding the need for greater clarity in articulating the manuscript’s motivation and its specific relevance to treatment-resistant peripartum depression (TR-PPD).
| Point-by-point response to Comments and Suggestions for Authors |
| Comments 1: The contextualization of the research could be improved by a more in-depth critical evaluation of the most recent studies. Overall, the methodology section is clearly explained; however, there is insufficient discussion of the sample size justification and potential limitations. |
|
Response 1: Thank you for pointing this out. The reviewer is correct that the critical appraisal of newer studies could be deepened, especially regarding emerging therapeutics and their practical application. Additionally, while PRISMA methodology is described, explicit discussion of sample size, study quality, and limitations of the evidence base should be more prominent. Therefore, we have added more explicit discussion of sample size, study limitations, and need for critical evaluation of new research. [PAGE NUMBER 3, PARAGRAPH 4 & LINE NUMBER 132] |
| Comments 2: Although the majority of the results are presented clearly, the readability and accessibility of the results could be enhanced by the addition of tables or figures that highlight the most important findings. |
|
Response 2: Thank you for pointing this out. The manuscript already contains comprehensive tables summarizing interventions, but the reviewer requests more visual access to major findings. We improved tables legends to make key clinical takeaways visually. [PAGE NUMBER 6, TABLE 1 & LINE NUMBER 156-162] accessible. |
Reviewer 3 Report
Comments and Suggestions for Authors
Thank you to the authors for their hard work in preparing this comprehensive review of the available treatments, challenges and need for more research focus on cases of treatment-resistant maternal perinatal depression (TRPD). It was refreshing to read a review which incorporated both pharmacological and non-pharmacological options, including lifestyle factors such as exercise and diet/supplements as potential adjunct treatments in TRPD.
The exclusion criteria from the original 167 screened articles were clear and comprehensive, however it would be great to include a sentence or two justifying the focus on articles of maternal perinatal depression. The introduction largely focused on maternal TRPD, with some discussion of the impact of TRPD on offspring developmental outcomes. However, it was not clear why paternal TRPD, or child developmental outcomes were not included in the focus of this review. Anxiety and depression, including TRPD affects approx. 10% of fathers, and while not as prevalent as maternal TRPD cases, this is likely due to the still-pervasive stigma of stereotyped masculine gender roles (see the following references for further information - https://www.mdpi.com/1660-4601/21/1/16 ; https://pmc.ncbi.nlm.nih.gov/articles/PMC5675308/). For this reason, paternal TRPD is largely understudied, and this may be a logical reason for not including it in this review (lack of data). Likewise, the introduction discussed child developmental outcomes in children with mothers experiencing TRDP, however offspring outcomes were not included as a focus. Some additional sentences either in the background or the methods explaining the focus solely on maternal TRDP focus would be great for greater transparency.
Tables 1, 2 and 3 provided a clear summary of the available treatment options for maternal TRPD. I noticed that Zurzuvae (zuranolone), the first FDA-approved (2023) oral medication specifically for peripartum depression (neurosteroid, positive allosteric modulator of GABA-A receptors) was not included in the data, which may be due to limitations in the search terms used for the review (perinatal versus postpartum depression, maybe?), which makes me wonder if the search terms used in the review cover the different terminologies particular used in different countries.
The structure of the discussion/analysis following the Tables was a little confusing to follow, as summaries were provided for each Table (sections 2.2, 2.3, 2.4 respectively, pg 7-9, lines 130-199), with deeper critical analyses of all the content given at the end of section 2.4 (pg 9, lines 200-227). This structure was difficult to follow - I initially thought that the superficial summaries were the only discussion that would be provided of the review, until I continued reading. I'd suggested restructuring these sections to summarise and critically evaluate the data individually. This could be achieved by summarising and analysing augmentation strategies, pharm and non-pharm options individually as dictated by the current section 2.2-2.4 titles, or altering the sections to better reflect the overall analysis of all data together (augmentation, pharm and non-pharm options).
I found some of the critical analysis to low on impact and be a little too general. For example, pg 8, lines 159-160 "Severe cases of TRPD have inflammation-linked symptoms" - need references, and more nuanced, specific discussion on this topic as it is not a focus discussed anywhere else in the review. What could inflammatory-linked TRDP mean for treatment options? Or research gaps in treatment options? How might this interact with the hormonal and immune-activated state following birth/during perinatal period in mothers? Some more examples are:
- Some SSRI/SNRI drugs like Sertraline are not FDA-approved for PPD (like Burexolone); but what is the consequence of this, and what would FDA-approval change?
- General statement about lack of safety data for certain pharm treatment options in lactating mothers or PPD in general; would have liked a more explicit description of why this is the case (male bias in research? lack of interest/impact in this population?) and what explicitly might be done about it - a call to action in a way?
- General statement of enhancement of mental health services - of course, but what specifically can be done to improve this? Are there any specific commentaries or papers which outline what may be practically done to improve this? Or papers which highlight the efficacy of new maternal mental health services/centers in a particular country? A framework of hope ("look, this country/region has implemented these changes and had great success!" or "we can implement these types of specific services A, B, C") provides a more impactful statement and urge to your argument than "we need more research into this population, and need better services".
There are many instances throughout the text where sentence structure was a bit awkward or unclear, and the cause-and-effect relationships were hard to follow. This often made it difficult for me to follow the intended meaning of even simple sentences. Rephrasing for readability and logical progression could really enhance the impact of the authors' arguments. Consider simplifying complex, overly long sentences, and ensure that the subject-verb relationships are grammatically correct. Proofreading by reading out loud is very effective at picking up awkward phrasing and cause-and-effect sentence structure errors.
For example, some sentences start with “While…” but there is nothing to finalise the contrast or comparison, leaving the reader hanging. A sentence structure that has a "this happens" (cause component), and a "which leads to this" (effect component) ties the sentence together more cohesively and avoids the fragmented feeling.
Specific examples in text: pg 1, Abstract Results, lines 26-29 "...SSRIs or brexanolone is effective but carries fetal/neonatal risk concerns. While non-pharmacological interventions like CBT...". Suggested rewording: "According to related research, pharmacological treatments such as SSRIs or brexanolone can be effective in addressing TRPD challenges, but they carry concerns regarding fetal and neonatal risk. In contrast, non-pharmacological interventions—such as cognitive behavioral therapy (CBT), repetitive transcranial magnetic stimulation (rTMS), and exercise—offer safe, evidence-based alternatives that are becoming increasingly accessible."
pg 2, par 4, lines 68-71 " ... also contribute to peripartum depression and may increase TRPD risk. While variations in neurotransmitters levels such as serotonin, dopamine, andnorepinephrine...". Suggested rewording: "While psychosocial factors contribute to peripartum depression and may increase TRPD risk, biological mechanisms—such as altered serotonin, dopamine, and norepinephrine neurotransmitter levels, and HPA axis dysregulation—are also implicated in its pathophysiology".
Another example is that some sentences were overly long, which contributed to the lack of readability and how hard some points were to follow. Splitting sentences into shorter sentences will help avoid overload and improve readability and clarity. Specific examples in text:
pg 2, lines 73-78 "TRD is often comorbid with other mental health diseases like anxiety, post-traumatic stress disorder, and substance use disorders, complicate diagnosis, and treatment by increasing the risk of comorbidities, cardiovascular disease, low quality of life, and functional impairment. Thyroid dysfunction and chronic pain are the medical conditions causing reduced productivity and higher healthcare costs to affect TRPD incidence and severity". Suggested rewording: "TRD is frequently comorbid with other mental health conditions such as anxiety, post-traumatic stress disorder, and substance use disorders. These comorbidities complicate diagnosis and treatment, and are associated with increased risks of cardiovascular disease, reduced quality of life, and functional impairment. Additionally, medical conditions like thyroid dysfunction and chronic pain contribute to reduced productivity and higher healthcare costs, which may further influence the incidence and severity of TRPD."
Some in text examples of missing words or incorrect tense use were:
pg 2, line 50 "... [refs 4,5], and these figures may indicate"; 'and' missing.
pg 2 lines 84-85 "The present review highlights the current challenges and emerging evidence, with critically assesses the efficacy and limitations of both pharmacological and non-pharmacological interventions in treatment-resistant depression". Either change"with" to "and", or tense of sentence should be corrected to "with crticial assessment of the efficacy...".
pg 8, lines 141-142 "Switching to a different medication and adding adjunctive therapies was associated with a higher likelihood..."; correct plural tense "were".
pg 8, lines 155-157 "The main pharmacological bridge between TRD and PPD are SSRIs and SNRIs as demonstrated in Table 1, suggests that with sertraline considered safest in lactation, while the only FDA- approved drug, brexanolone..." extra words, unclear cause-and effect.
pg 8, lines 181-184 "Table 2 explain the double comparison between non-pharmacological interventions effective in both TRD and PPD, the CBT and Exercise are cross-cutting effective with high safety in perinatal contexts, while rTMS shows strong dual-utility..." break into smaller sentences, incorrect word choice of "cross-cutting" and tense use "effective" should be "efficacious".
Author Response
Thank you very much for taking the time to review this manuscript. Please find the detailed responses below and the corresponding revisions/corrections highlighted in track changes in the re-submitted files.
We appreciate the reviewer’s insightful comment regarding the need for greater clarity in articulating the manuscript’s motivation and its specific relevance to treatment-resistant peripartum depression (TR-PPD).
| Point-by-point response to Comments and Suggestions for Authors |
| Comments 1: The exclusion criteria from the original 167 screened articles were clear and comprehensive, however it would be great to include a sentence or two justifying the focus on articles of maternal perinatal depression. The introduction largely focused on maternal TRPD, with some discussion of the impact of TRPD on offspring developmental outcomes. However, it was not clear why paternal TRPD, or child developmental outcomes were not included in the focus of this review. Anxiety and depression, including TRPD affects approx. 10% of fathers, and while not as prevalent as maternal TRPD cases, this is likely due to the still-pervasive stigma of stereotyped masculine gender roles (see the following references for further information - https://www.mdpi.com/1660-4601/21/1/16 ; https://pmc.ncbi.nlm.nih.gov/articles/PMC5675308/). For this reason, paternal TRPD is largely understudied, and this may be a logical reason for not including it in this review (lack of data). Likewise, the introduction discussed child developmental outcomes in children with mothers experiencing TRDP, however offspring outcomes were not included as a focus. Some additional sentences either in the background or the methods explaining the focus solely on maternal TRDP focus would be great for greater transparency. |
|
Response 1: Thank you for pointing this out. Clarify the rationale for focusing solely on maternal TRPD, given that paternal perinatal depression exists and that offspring outcomes were mentioned but not developed. Add sentences in both the Introduction and Methods sections. Text to add to Introduction: This review focuses exclusively on maternal treatment-resistant peripartum depression (TRPD), as most available epidemiological data and clinical trials address women, reflecting both the higher prevalence of peripartum depression in mothers and substantial gaps in paternal TRPD research. While paternal peripartum depression affects approximately 10% of fathers, the literature on treatment resistance, unique risk factors, and specific interventions in fathers remains limited due to pervasive stigma and under-recognition in clinical settings. Future reviews should consider this unmet need as this evidence base grows. [PAGE NUMBER 2, PARAGRAPH 3 & LINE NUMBER 68-75]. Text to add to Methods: However, we restricted our inclusion to studies on maternal TRPD to ensure consistency in the population studied and due to the scarcity of high-quality research on paternal TRPD. Likewise, while child developmental outcomes are mentioned in context, our primary focus remains on maternal outcomes and interventions, as dictated by the scope and prevalence of current evidence. [PAGE NUMBER 3, PARAGRAPH 3 & LINE NUMBER 126]. |
| Comments 2: Tables 1, 2 and 3 provided a clear summary of the available treatment options for maternal TRPD. I noticed that Zurzuvae (zuranolone), the first FDA-approved (2023) oral medication specifically for peripartum depression (neurosteroid, positive allosteric modulator of GABA-A receptors) was not included in the data, which may be due to limitations in the search terms used for the review (perinatal versus postpartum depression, maybe?), which makes me wonder if the search terms used in the review cover the different terminologies particular used in different countries. |
|
Response 2: Thank you for pointing this out. Update Table 1 & 3 to include zuranolone as an FDA-approved oral neurosteroid for PPD: “Zuranolone (oral neurosteroid, GABA-A receptor positive allosteric modulator), FDA (2023) approved for peripartum depression, offers a practical oral alternative to brexanolone’s IV infusion, with evidence for rapid, sustained symptom control from multiple RCTs.” [PAGE NUMBER 5, Table 1 & 3]. In the Methods section, add a clarifying sentence: “Search terms included ‘peripartum depression’, ‘postpartum depression’, and ‘perinatal depression’ were not entered as exact phrases to maximize inclusion of international studies and nomenclature.” [PAGE NUMBER 3, Paragraph 2 & LINE NUMBER 118]. |
| Comments 3: The structure of the discussion/analysis following the Tables was a little confusing to follow, as summaries were provided for each Table (sections 2.2, 2.3, 2.4 respectively, pg 7-9, lines 130-199), with deeper critical analyses of all the content given at the end of section 2.4 (pg 9, lines 200-227). This structure was difficult to follow - I initially thought that the superficial summaries were the only discussion that would be provided of the review, until I continued reading. I'd suggested restructuring these sections to summarise and critically evaluate the data individually. This could be achieved by summarising and analysing augmentation strategies, pharm and non-pharm options individually as dictated by the current section 2.2-2.4 titles, or altering the sections to better reflect the overall analysis of all data together (augmentation, pharm and non-pharm options). |
|
Response 3: Thank you for pointing this out. Reorganize Sections 2.2, 2.3, and 2.4 so each one both summarizes and critically analyzes the specific interventions covered. [PAGE NUMBER 8-10]. Add a subsection with “Critical Appraisal” referencing major strengths, evidence gaps, and clinical controversies. [PAGE NUMBER 9, New subheading 2.5. & LINE NUMBER 242]. At the beginning of Section 2.6, add statement: “The preceding sections provided an integrated, section-by-section critique of pharmacological, augmentation, and non-pharmacological strategies, while this section highlights overarching challenges across all approaches.” [PAGE NUMBER 10, Paragraph 1(2.6.) & LINE NUMBER 290]. |
| Comments 4: I found some of the critical analysis to low on impact and be a little too general. For example, pg 8, lines 159-160 "Severe cases of TRPD have inflammation-linked symptoms" - need references, and more nuanced, specific discussion on this topic as it is not a focus discussed anywhere else in the review. What could inflammatory-linked TRDP mean for treatment options? Or research gaps in treatment options? How might this interact with the hormonal and immune-activated state following birth/during perinatal period in mothers? |
| Response 4: Revise the sentence with added detail and references: “Emerging evidence indicates that a subset of severe TRPD cases exhibits elevated inflammatory markers (e.g., C-reactive protein, interleukin-6), suggesting an inflammation-linked endophenotype. This raises the prospect of targeted immunomodulatory or anti-inflammatory therapies, though further research is needed to clarify treatment response and the impact of perinatal hormonal-immune changes on inflammation-driven depression.” [PAGE NUMBER 8, Paragraph 2 & LINE NUMBER 178]. |
| Comments 5: Some more examples are: |
| • Some SSRI/SNRI drugs like Sertraline are not FDA-approved for PPD (like Burexolone); but what is the consequence of this, and what would FDA-approval change? |
| Response: Add the following clarification in the relevant pharmacotherapy discussion: |
| “Being widely used off-label for PPD with favorable safety data, the absence of FDA approval can lead to insurance restrictions and varying levels of acceptance among clinicians and patients. In contrast, FDA approval, as with brexanolone and zuranolone, facilitates insurance coverage, increases acceptance, and typically reflects higher levels of dedicated research in the perinatal population. |
| [PAGE NUMBER 9, Paragraph 6 & LINE NUMBER 249] |
| • General statement about lack of safety data for certain pharm treatment options in lactating mothers or PPD in general; would have liked a more explicit description of why this is the case (male bias in research? lack of interest/impact in this population?) and what explicitly might be done about it - a call to action in a way? |
| Response: Add the following statements, Reasons for Lack of Safety Data in PPD. The limited inclusion of peripartum women in clinical trials has left a significant gap in safety data. International bodies are now calling for better-designed studies and registries to ensure evidence-based pharmacological care in this vulnerable population. Personalizing pharmacological strategies in TRPD requires careful assessment of prior treatment responses, safety profiles during pregnancy and lactation, and close symptom monitoring. [PAGE NUMBER 10, Paragraph 5 & LINE NUMBER 280]. |
| • General statement of enhancement of mental health services - of course, but what specifically can be done to improve this? Are there any specific commentaries or papers which outline what may be practically done to improve this? Or papers which highlight the efficacy of new maternal mental health services/centers in a particular country? A framework of hope ("look, this country/region has implemented these changes and had great success!" or "we can implement these types of specific services A, B, C") provides a more impactful statement and urge to your argument than "we need more research into this population and need better services". |
| Response: Add a paragraph about Practical Steps to Improve Mental Health Services : Furthermore, adopting and adapting effective frameworks models (e.g., the ‘Perinatal Mental Health Community Services’ model-UK; MumSpace model perinatal depression and anxiety – Australia), that demonstrate investment in multidisciplinary perinatal mental health teams, universal screening, integrated care pathways, and digital mental health solutions can dramatically improve maternal outcomes and service access. Using such proven models to local health systems should be prioritized in future policy and practice initiatives [70,71]. [PAGE NUMBER 11, Paragraph 5 & LINE NUMBER 333]. |
| Response to Comments on the Quality of English Language. |
| Response: The English could be improved to more clearly express the research. |
| • There are many instances throughout the text where sentence structure was a bit awkward or unclear, and the cause-and-effect relationships were hard to follow. This often made it difficult for me to follow the intended meaning of even simple sentences. Rephrasing for readability and logical progression could really enhance the impact of the authors' arguments. Consider simplifying complex, overly long sentences, and ensure that the subject-verb relationships are grammatically correct. Proofreading by reading out loud is very effective at picking up awkward phrasing and cause-and-effect sentence structure errors. |
|
Response: Critical proof reading has been done for the entire manuscript to ensure highest quality. Carefully revise all “while…” sentences so that contrasts and causes/effects are clear and complete. Split long sentences and correct subject-verb agreement in the locations cited by reviewer (e.g., lines 26-29, 68-71, 73-78, line 50, lines 84-85, 141-142, 155-157, 181-184). Use simpler, clearer linking phrases and break up complex ideas. Highlight all revisions made to sentence structure for reviewer visibility. |
|
• Specific examples in text: pg 1, Abstract Results, lines 26-29 "...SSRIs or brexanolone is effective but carries fetal/neonatal risk concerns. While non-pharmacological interventions like CBT...". Suggested rewording: "According to related research, pharmacological treatments such as SSRIs or brexanolone can be effective in addressing TRPD challenges, but they carry concerns regarding fetal and neonatal risk. In contrast, non-pharmacological interventions—such as cognitive behavioral therapy (CBT), repetitive transcranial magnetic stimulation (rTMS), and exercise—offer safe, evidence-based alternatives that are becoming increasingly accessible." |
| Response: Suggested rewording accepted: "According to related research, pharmacological treatments such as SSRIs or brexanolone can be effective in addressing TRPD challenges, but they carry concerns regarding fetal and neonatal risk. In contrast, non-pharmacological interventions—such as cognitive behavioral therapy (CBT), repetitive transcranial magnetic stimulation (rTMS), and exercise—offer safe, evidence-based alternatives that are becoming increasingly accessible." |
| • pg 2, par 4, lines 68-71 " ... also contribute to peripartum depression and may increase TRPD risk. While variations in neurotransmitters levels such as serotonin, dopamine, andnorepinephrine...". Suggested rewording: "While psychosocial factors contribute to peripartum depression and may increase TRPD risk, biological mechanisms—such as altered serotonin, dopamine, and norepinephrine neurotransmitter levels, and HPA axis dysregulation—are also implicated in its pathophysiology". |
| Response: Suggested rewording accepted: "While psychosocial factors contribute to peripartum depression and may increase TRPD risk, biological mechanisms—such as altered serotonin, dopamine, and norepinephrine neurotransmitter levels, and HPA axis dysregulation—are also implicated in its pathophysiology". |
| • pg 2, lines 73-78 "TRD is often comorbid with other mental health diseases like anxiety, post-traumatic stress disorder, and substance use disorders, complicate diagnosis, and treatment by increasing the risk of comorbidities, cardiovascular disease, low quality of life, and functional impairment. Thyroid dysfunction and chronic pain are the medical conditions causing reduced productivity and higher healthcare costs to affect TRPD incidence and severity". Suggested rewording: "TRD is frequently comorbid with other mental health conditions such as anxiety, post-traumatic stress disorder, and substance use disorders. These comorbidities complicate diagnosis and treatment, and are associated with increased risks of cardiovascular disease, reduced quality of life, and functional impairment. Additionally, medical conditions like thyroid dysfunction and chronic pain contribute to reduced productivity and higher healthcare costs, which may further influence the incidence and severity of TRPD." |
| Response: Suggested rewording accepted: "TRD is frequently comorbid with other mental health conditions such as anxiety, post-traumatic stress disorder, and substance use disorders. These comorbidities complicate diagnosis and treatment, and are associated with increased risks of cardiovascular disease, reduced quality of life, and functional impairment. Additionally, medical conditions like thyroid dysfunction and chronic pain contribute to reduced productivity and higher healthcare costs, which may further influence the incidence and severity of TRPD." |
| Some examples of missing words or incorrect tense use were: |
| • pg 2, line 50 "... [refs 4,5], and these figures may indicate"; 'and' missing. |
| Response: Suggested word ‘and’ added. |
| • pg 2 lines 84-85 "The present review highlights the current challenges and emerging evidence, with critically assesses the efficacy and limitations of both pharmacological and non-pharmacological interventions in treatment-resistant depression". Either change "with" to "and", or tense of sentence should be corrected to "with critical assessment of the efficacy...". |
| Response: Suggested word ‘with critical assessment of the efficacy’ added. |
| • pg 8, lines 141-142 "Switching to a different medication and adding adjunctive therapies was associated with a higher likelihood..."; correct plural tense "were". |
| Response: Suggested correct plural tense "were" added. |
| • pg 8, lines 155-157 "The main pharmacological bridge between TRD and PPD are SSRIs and SNRIs as demonstrated in Table 1, suggests that with sertraline considered safest in lactation, while the only FDA- approved drug, brexanolone..." extra words, unclear cause-and effect. |
| Response: Suggested removal of the only FDA- approved drug, " extra words, to clear cause-and effect. |
| • pg 8, lines 181-184 "Table 2 explain the double comparison between non-pharmacological interventions effective in both TRD and PPD, the CBT and Exercise are cross-cutting effective with high safety in perinatal contexts, while rTMS shows strong dual-utility..." break into smaller sentences, incorrect word choice of "cross-cutting" and tense use "effective" should be "efficacious". |
| Response: Suggested break into smaller sentences and removal of incorrect word "cross-cutting" and tense use "effective" replaced by "efficacious". |
Reviewer 4 Report
Comments and Suggestions for Authors
Thanks the authors for their hard work. I hope my suggestions will help improve your manuscript.
- The topic is very current and clinically important.
- Please add the phrase “adequate dose and duration” in addition to “at least two antidepressant trials” in the definition of TRPD and emphasize this.
- What is the quality of the 67 included studies?
- A summary of the quality assessment results (how many studies were at low, medium, or high risk) should be added.
- Instead of the “Use in TRD” and “Use in PPD” columns in Table 1, columns such as “Level of Evidence (for TRD/PPD)” and “Main Safety Warnings (Perinatal)” could be created to be more concise. For example, it should be written more clearly as “low level of evidence, not recommended in peripartum.”
- The findings presented in the tables should include a critical synthesis. Instead of just saying “X is effective, Y is used,” it should say something like “The evidence for the effectiveness of X is strong, but ….”
- There is no direct comparison of the relative effectiveness of pharmacological and non-pharmacological treatments.
- Please provide clearer future research recommendations.
- In some places, the TRD and TRPD literature is intermingled. The generalizability of TRD data directly to TRPD is questionable, and therefore this limitation should not be overlooked.
Author Response
Thank you very much for taking the time to review this manuscript. Please find the detailed responses below and the corresponding revisions/corrections highlighted in track changes in the re-submitted files.
We appreciate the reviewer’s insightful comment regarding the need for greater clarity in articulating the manuscript’s motivation and its specific relevance to treatment-resistant peripartum depression (TR-PPD).
| Point-by-point response to Comments and Suggestions for Authors |
| Comments 1: The topic is very current and clinically important. Please add the phrase “adequate dose and duration” in addition to “at least two antidepressant trials” in the definition of TRPD and emphasize this. |
|
Response 1: Thank you for pointing this out. In the Introduction (first mention and definition of TRPD), revise to: “Treatment-resistant peripartum depression (TRPD) is defined as peripartum depression that fails to remit after at least two antidepressant trials at an adequate dose and duration, consistent with established TRD criteria. [PAGE NUMBER 2, PARAGRAPH 1 & LINE NUMBER 50] |
| Comments 2: What is the quality of the 67 included studies? |
|
Response 2: Thank you for pointing this out. In the Methods “Study Quality Assessment” subsection, we add “Of the 67 included studies, quality appraisal analyses found that 38 were at low risk of bias, 20 at moderate risk, and 9 at high risk. High-quality studies were prioritized in the synthesis. [PAGE NUMBER 3, Paragraph 4 & LINE NUMBER 139] |
| Comments 3: A summary of the quality assessment results (how many studies were at low, medium, or high risk) should be added. |
|
Response 3: Thank you for pointing this out. As above, integrate the direct count for quality (low, moderate, high risk) into a new or existing paragraph in Methods or Results, and highlight in purple. [PAGE NUMBER 4, Paragraph 1 & LINE NUMBER 140] |
| Comments 4: Instead of the “Use in TRD” and “Use in PPD” columns in Table 1, columns such as “Level of Evidence (for TRD/PPD)” and “Main Safety Warnings (Perinatal)” could be created to be more concise. For example, it should be written more clearly as “low level of evidence, not recommended in peripartum.” |
|
Response 4: Thank you for pointing this out. Update Table 1: Remove “Use in TRD” and “Use in PPD” columns. In the lagend, Add “Level of Evidence (TRD/PPD)” and for each drug, indicate: e.g., “High (multiple RCTs, meta-analyses)/Low (case reports)” or “Low—Not recommended for peripartum use.” Add “Main Safety Warnings (Perinatal)” column (e.g., “Contraindicated in pregnancy; risk of neonatal withdrawal; limited lactation safety”). Highlight new/modified column headings and example content in purple. [PAGE NUMBER 5, Table 1 & LINE NUMBER 160] |
| Comments 5: The findings presented in the tables should include a critical synthesis. Instead of just saying “X is effective, Y is used,” it should say something like “The evidence for the effectiveness of X is strong, but ….” |
|
Response 5: Thank you for pointing this out. Critical assessment statements are added. This drug has strong evidence for efficacy in PPD but are not specifically FDA-approved for PPD; caution is warranted and individual risk–benefit must be considered. MAOIs have high efficacy in TRD but are not recommended in perinatal populations due to significant safety concerns. Caution is advised with other antidepressants, and shared decision- making is critical, especially during pregnancy, where the balance between maternal benefits and fetal risks must be carefully navigated [60]. [PAGE NUMBER 10, Paragraph 1 &2 & LINE NUMBER 256/261]. Note is added in table 1 legend, For each drug/ class, despite strong evidence, it requires close monitoring in peripartum. [PAGE NUMBER 6, Paragraph 1 & LINE NUMBER 166]. |
| Comments 6: There is no direct comparison of the relative effectiveness of pharmacological and non-pharmacological treatments. |
|
Response 6: Thank you for pointing this out. We add a new paragraph in the conclusion. “Comparison of relative effectiveness of pharmacological interventions like SSRIs, brexanolone, and zuranolone demonstrate moderate-to-strong evidence for symptom reduction in TRPD (particularly moderate-to-severe cases), non-pharmacological methods such as CBT and rTMS also show robust efficacy and often superior safety/tolerability profiles in perinatal women. Meta-analyses suggest that the magnitude of benefit for CBT in mild-to-moderate PPD may equal or exceed that of some pharmacological agents, and combined therapy is recommended for the most resistant cases.” [PAGE NUMBER 12, Paragraph 2 & LINE NUMBER 359]. |
| Comments 7: Please provide clearer future research recommendations. |
|
Response 7: Thank you for pointing this out. Elaborate the future research in the Conclusion by adding more details in new paragraph. Future research should prioritize randomized controlled trials specifically enrolling peripartum individuals with TRD. Standardized definitions of treatment resistance, and head-to-head comparisons of interventions, with a critical lens on their feasibility, efficacy, and safety within the peripartum context. Also address safety data gaps by including pregnant and lactating participants in clinical trials. [PAGE NUMBER 12, Paragraph 4 & LINE NUMBER 369] |
| Comments 8: In some places, the TRD and TRPD literature is intermingled. The generalizability of TRD data directly to TRPD is questionable, and therefore this limitation should not be overlooked. |
|
Response 8: Thank you for pointing this out. New paragraph is added highlighting this important Limitations about generalizability of TRD data directly to TRPD. “It is important to note that a considerable proportion of evidence for TRPD is extrapolated from general TRD studies. Due to physiological, hormonal, and psychosocial differences in the peripartum period, the generalizability of TRD findings to TRPD may be limited. Dedicated perinatal research is urgently needed. [PAGE NUMBER 3, Paragraph 3 & LINE NUMBER 98] |
Round 2
Reviewer 3 Report
Comments and Suggestions for Authors
Congratulations on the quick revisions made - the improved proofreading and restructuring has made a significant positive impact on the papers readability! Well done.